# Applying High-Resolution Satellite and UAS Imagery for Detecting Coldwater Inputs in Temperate Streams of the Iowa Driftless Region

**Niti B. Mishra** [1,2,*], **Michael J. Siepker** [3] **and Greg Simmons** [4]

1 Department of Geography & Environmental Science, University of Wisconsin-La Crosse, La Crosse, WI 54601, USA
2 River Studies Center, University of Wisconsin-La Crosse, La Crosse, WI 54601, USA
3 Iowa Department of Natural Resources, Decorah, IA 52101, USA; michael.siepker@dnr.iowa.gov
4 Iowa Department of Natural Resources, 22693 205th Ave, Manchester, IA 52057, USA; greg.simmons@dnr.iowa.gov
* Correspondence: nmishra@uwlax.edu; Tel.: +1-608-785-8334

**Abstract:** Coldwater streams are crucial habitats for many biota including Salmonidae and Cottidae species that are unable to tolerate warmer water temperatures. Accurate classification of coldwater streams is essential for their conservation, restoration, and management, especially in light of increasing human disturbance and climate change. Coldwater streams receive cooler groundwater inputs and, as a result, typically remain ice-free during the winter. Based on this empirical thermal evidence, we examined the potential of very high-resolution (VHR) satellite and uncrewed aerial system (UAS) imagery to (i) detect coldwater streams using semi-automatic classification versus visual interpretation approaches, (ii) examine the physical factors that contribute to inaccuracies in detecting coldwater habitats, and (iii) use the results to identify inaccuracies in existing thermal stream classification datasets and recommend coverage updates. Due to complex site conditions, semi-automated classification was time consuming and produced low mapping accuracy, while visual interpretation produced better results. VHR imagery detected only the highest quality coldwater streams while lower quality streams that still met the thermal and biological criteria to be classified as coldwater remained undetected. Complex stream and site variables (narrow stream width, canopy cover, terrain shadow, stream covered by ice and drifting snow), image quality (spatial resolution, solar elevation angle), and environmental conditions (ambient temperature prior to image acquisition) make coldwater detection challenging; however, UAS imagery is uniquely suited for mapping very narrow streams and can bridge the gap between field data and satellite imagery. Field-collected water temperatures and stream habitat and fish community inventories may be necessary to overcome these challenges and allow validation of remote sensing results. We detected >30 km of coldwater streams that are currently misclassified as warmwater. Overall, visual interpretation of VHR imagery it is a relatively quick and inexpensive approach to detect the location and extent of coldwater stream resources and could be used to develop field monitoring programs to confirm location and extent of coldwater aquatic resources.

**Keywords:** coldwater streams; stream temperature; satellite imagery; UAS imagery; surface water—ground water interactions; geographic object-based image analysis

## 1. Introduction

Water temperature is a crucial determinant influencing all trophic levels in rivers and streams [1]. The surface water temperature and thermal regime are often modulated by their interaction with groundwater over space and time [2]. In northern temperate climates, groundwater inputs to flowing waters are usually warmer than surface water in winter [3] and cooler in summer [4]. Streams with relatively cold and stable summer water

temperatures below 68 °F (20 °C) are considered coldwater streams [5,6], and are often dominated by stenothermic Salmonidae and Cottidae species that are unable to tolerate warm water temperatures [5,7,8]. Many coldwater streams have been altered by human activities (e.g., land use changes, water withdrawals, and pollution) [9,10] and could be vulnerable to the impacts of climate change, including changes in water temperature and flow patterns [8,11].

The conservation, restoration, and management of coldwater streams requires that managers and decision makers first accurately know where coldwater habitats exist in the landscape. A variety of tools and techniques have been utilized for detecting coldwater streams. For example, in situ stream fish surveys involving direct observation and sampling to determine the presence of coldwater fishes have been extensively utilized [12]. Additionally, continuous water temperature loggers have been used to identify coldwater habitats and monitor temporal changes in water temperatures [13–15]. However, field-based monitoring providing accurate data on a single location requires extensive sampling efforts when attempting to characterize groundwater inputs and stream conditions over large geographical areas.

Remotely sensed imagery has served as a useful tool for identifying and monitoring coldwater streams. In particular, orbital satellite sensors acquiring imagery in the thermal infrared (TIR) wavelength of the electromagnetic spectrum have been used to map stream temperatures [16,17]. While satellite-acquired thermal imagery is able to map temperatures of wider streams (i.e., stream width > 100 m) [18–20], its effectiveness for monitoring narrower streams is limited due to coarse spatial resolution (i.e., pixel size > 50 m) [21,22]. Additionally, orbital sensors are less sensitive to fine-scale variation in water temperature and their magnitude of error could be relatively high, making accurate temperature mapping challenging [23]. To overcome these limitations, studies have utilized uncrewed aerial system (UAS)-mounted TIR sensors to map stream temperatures with higher accuracy [24–26] and successfully detect cold groundwater inputs on narrow coldwater stream segments [27,28]. Although UAS-mounted TIR sensors provide spatially explicit and relatively accurate stream temperature maps, UAS-based mapping has practical limitations and is not suitable for watershed to regional scale monitoring of streams [24,29].

Orbital very high-resolution (VHR) TIR imagery (pixel size $\leq$ 2 m) is currently not available [19], but optical VHR imagery (i.e., acquired in the visible wavelength of light) is widely available and has been extensively utilized for detecting and mapping rivers and streams [30], assessing the impacts of restoration efforts on stream geomorphic properties [18,31], characterizing submerged and emergent aquatic vegetation [32–34], and detecting changes in stream water quality [35–37]. While optical VHR imagery does not enable the direct estimation of stream temperature, recent studies have found that the high spatial detail of these images could be used to interpret stream conditions and potentially enable the identification of coldwater habitats [16].

Generally, surface waters that receive substantial groundwater inputs have less variable water temperatures compared to those that are not groundwater-controlled [38]. For example, groundwater entering streams in the winter is significantly warmer (approx. 52 °F) than the surrounding surface water temperatures, allowing the streams to remain mostly ice-free. Streams without this groundwater input are colder and mostly ice-covered during winter [16,38,39]. We found this empirical observation to be true in our study area as well (Figure 1).

Perennial surface waters in Iowa are currently classified thermally as warmwater or coldwater [40]. Briefly, waters are considered warmwater unless their temperature and flow maintain a variety of coldwater species, including populations of trout and/or associated aquatic communities. Maximum daily water temperatures not exceeding 24 °C, presence of watercress (*Nasturtium officinale*), populations of sculpin (*Cottus* spp.), and self-sustaining populations of trout are just a few of the indicators that support the designation of stream segments as coldwater. Since the initial classification of coldwater stream segments in Iowa was supported by limited field data, extensive new data sets on Iowa streams provide

the opportunity to revise the list of streams supporting coldwater communities. If stream segments are currently classified as warmwater, but new data supports reclassification to coldwater, the available data can be reviewed by the State of Iowa and the classification updated where appropriate.

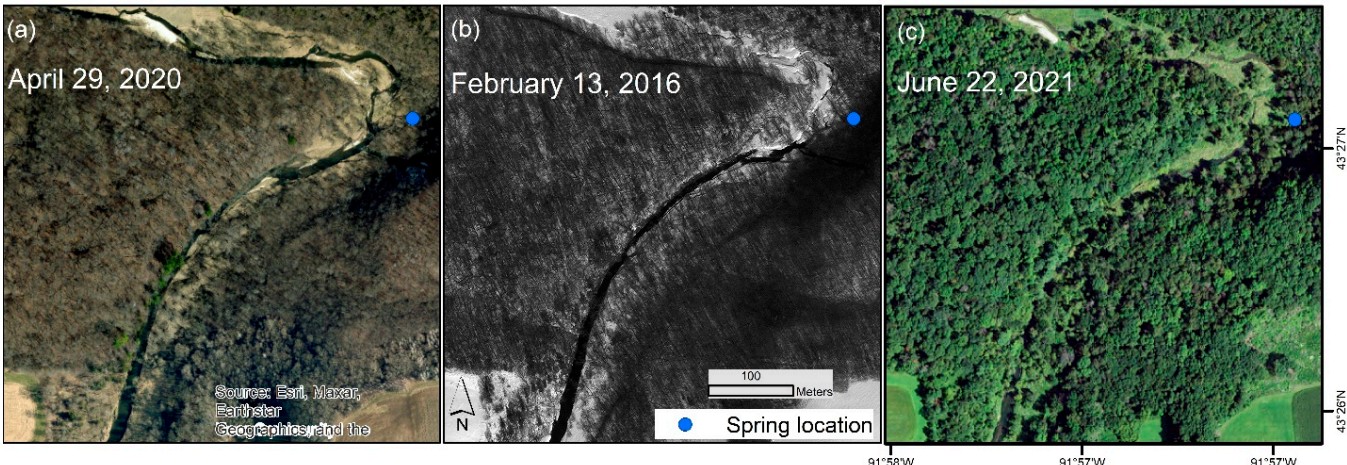

**Figure 1.** Satellite imagery of a northeast Iowa stream taken during spring (**a**), winter (**b**), and summer (**c**) showing the location of groundwater input into the stream. Regardless of thermal condition, both stream segments appear identical in the spring and summer imagery (**a,c**). The winter imagery (**b**) shows the segment of stream influenced by groundwater. The stream segment downstream of the groundwater input remains open while the segment upstream of the groundwater input is ice- and snow-covered. (Image copyright 2023 Maxar).

Therefore, the objectives of our study were to (i) use optical VHR imagery collected during winter seasons to delineate open water reaches of streams in a northeastern Iowa watershed using semi-automatic and manual approaches, (ii) validate the coldwater stream status of selected reaches using water temperature data, fish community data, instream habitat data, and UAS imagery, and (iii) use the extent of coldwater stream segments identified in this study to recommend updates to existing stream classification datasets. We also discuss the physical factors that contribute to inaccuracies in detecting coldwater habitats. Finally, we discuss how our information could be used by managers and policy makers to more accurately document coldwater stream resources in Iowa and beyond.

## 2. Materials and Methods

### 2.1. Study Area

The Canoe Creek watershed lies within the Paleozoic Plateau ecoregion, also known as the Driftless Region, of northeastern Iowa, USA (Figure 2). Canoe Creek is a HUC 10 watershed that covers approximately 172 km², ranges in elevation from 219 m to 419 m, and contains roughly 733 km of streams including Canoe Creek, North Canoe Creek, and their tributaries [41]. Land use and cover within the watershed is predominantly pasture and cropland with tracts of deciduous forest [42]. The karst topography that dominates this region creates an eroded valley-ridge patchwork with limestone and dolomite rock out-croppings [43] and an extensive network of coldwater streams, providing suitable habitat for coldwater fishes [44]. Coldwater streams in the area are maintained by ephemeral and perennial baseflow that enters the streams from numerous discreet springs and seeps found throughout the watersheds. As a result, stream water temperatures can be quite variable, ranging from warmwater to coldwater conditions throughout the watershed, depending on the level of groundwater inputs. These coldwater streams are able to support a variety of flora and fauna not found in other parts of Iowa [45]. Fishes found within coldwater stream segments can include native slimy sculpin (*Cottus cognatus*), mottled sculpin (*Cottus bairdii*), brook trout (*Salvelinus fontinalis*), and introduced brown trout (*Salmo trutta*), whereas the

warmer stream segments support typical Midwestern USA warmwater stream communities. Seasonal water temperatures and fish community structure have been monitored at various locations throughout the watershed as part of the Iowa Department of Natural Resources' (IA DNR) biological monitoring.

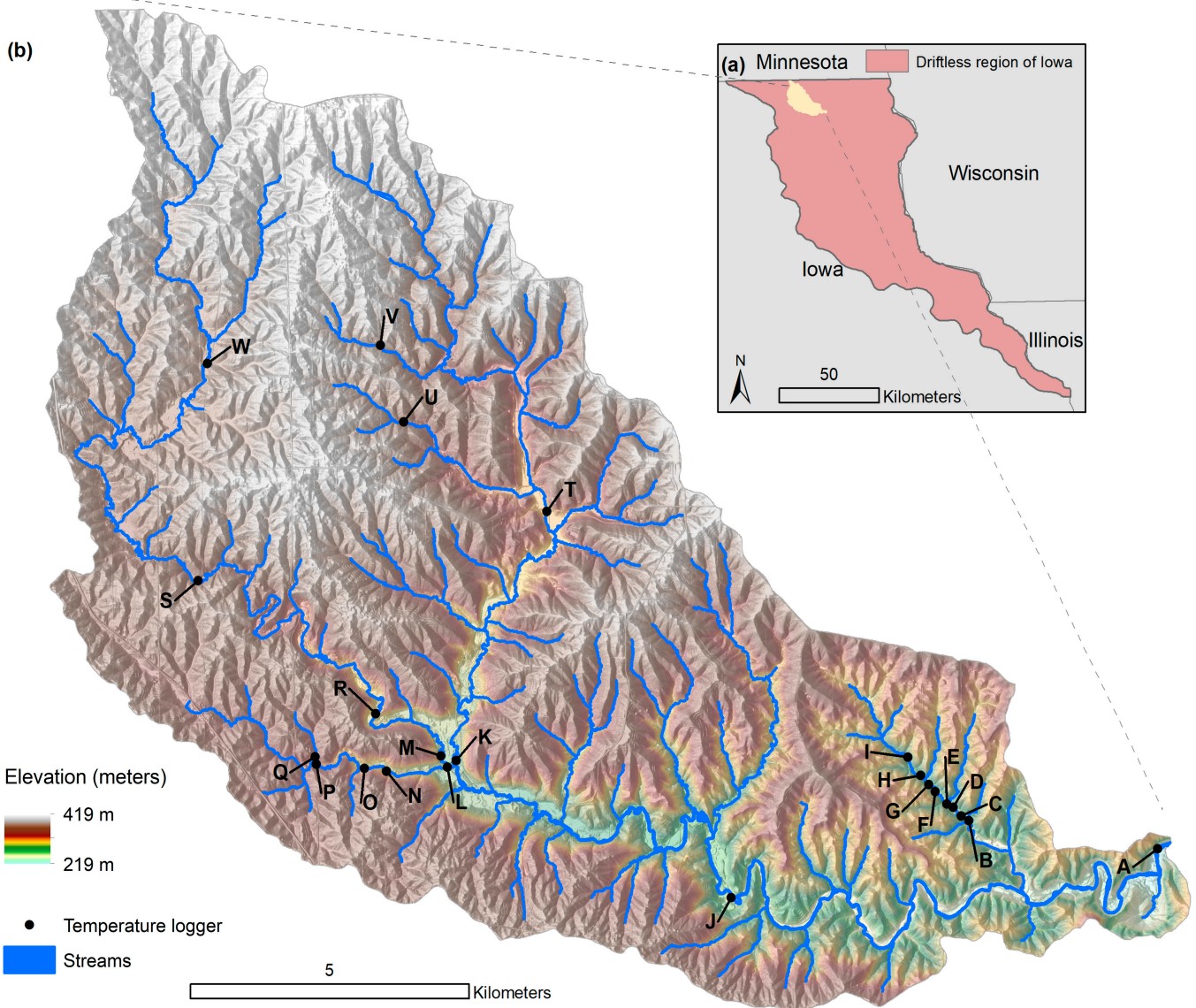

**Figure 2.** (**a**) Location of the Canoe Creek watershed within the Driftless Region of Iowa; (**b**) streams in the study watershed and locations where continuous water temperature data (black dot) was available. The watershed is shown using hillshade coverage. Site IDs (A–W) for each water temperature monitoring location are also shown.

### 2.2. Water Temperature Assessments

In situ continuous water temperature data was available at 23 locations in the Canoe Creek watershed (Figure 2). Most water temperature data was collected from 2018 to present; however, some data was collected as early as 2010. Data loggers (HOBO 64K Pendant; Onset Computer Corp., Bourne, MA, USA) were set to record the hourly stream temperature from spring to autumn of the survey year. Water temperature data collected during 2021 was not included in the analysis since the area experienced severe drought conditions during summer of that year [46].

### 2.3. Fish Community Assessments

Fish community data from several sources was used in this evaluation, including targeted coldwater stream surveys. Prior to targeted coldwater field data collection, winter satellite imagery (Maxar Technologies, Inc. Westminster, CO, USA) was used to select stream locations that had a high probability of being coldwater habitats. Relatively long sections of open water, visible from the winter imagery, that had limited sampling history were given priority for evaluation. Fishery field surveys were conducted at twelve new sites in 2020. Another nine sites that included additional fish community data (2010 to present) from various Iowa DNR biological monitoring projects were also included in the analysis to further validate imagery interpretation (Figure 3). From May to September 2020, stream fishes and habitat were sampled following Iowa DNR wadeable stream sampling methodologies [40]. Briefly, three stream reaches were sampled at each survey site, with reach length set at 40 mean stream widths. Depending on the stream width, fish communities were assessed with a single upstream sampling pass using a single backpack (wetted width < 5 m), tandem backpack (wetted width > 5 m), or barge (wetted width > 5 m and average depth > 0.5 m) electrofisher. All captured fishes were identified and enumerated; electrofishing time was recorded for calculation of catch per unit effort (CPUE). Total lengths (mm) of all trout were also measured. Instream physical habitat was also characterized for each sample reach. Water temperature, conductivity, wetted stream width, and stream discharge were also measured, as was the presence and abundance of watercress at each sample reach.

### 2.4. Uncrewed Aerial System (UAS) Survey

To better understand the geomorphometric properties of coldwater streams, we collected UAS imagery at four selected sites (i.e., sites 1, 2, 4, and 8 (shown in Figure 3)) on a single day (31 August 2021). These sites were selected to represent various stream conditions (e.g., stream width, canopy cover, relief/terrain variation). Images were collected using a rotary-wing UAS (Mavic 2 pro from DJI) fitted with a GPS/GNSS satellite-positioning system and a 20-megapixel Hasselblad camera (i.e., 5472 by 3648 pixels) that captures JPEG images. Map Pilot for the DJI app was used to pre-program the mission parameters, which were uploaded to the UAS autopilot, which was instructed to fly in a grid pattern at a constant elevation (with respect to the ground) using the 'terrain follow' feature. The image acquisition consisted of one flight at each location. For all flights, the average flight altitude was set to 90 m above the ground, the forward image overlap to 80%, the side overlap to 75%, and the flight speed to 4 m s$^{-1}$. Images were processed using the structure from motion principles in the Pix4D mapper pro software to produce orthomosaic and digital elevation models for each site.

### 2.5. Image Selection, Pre-Processing, and Analysis

A workflow of the methodology followed in this study is shown in Figure 4. We searched DigitalGlobe's (Maxar Technologies, Inc. Westminster, CO, USA) optical VHR imagery archive from 2010 to 2020 for images taken during the winter season. We then filtered the available imagery for only images acquired during winter periods when the ground was covered by snow and temperatures did not exceed 0 °C for at least four days prior to image acquisition (Figure 3) to ensure melt water did not influence image interpretation. Two panchromatic images (24 February 2014 and 13 February 2016) and one multispectral image (13 February 2016) met these requirements and were selected for further interpretation. The 2014 imagery covered the entire study watershed whereas the 2016 imagery had approximately 80% coverage (Figure 4). Visual comparison of multispectral (2 m spatial resolution) versus panchromatic (0.5 m spatial resolution) imagery revealed that streams with widths <8 m were not clearly identifiable in the multispectral image but were clearly visible in the panchromatic imagery. Therefore, panchromatic imagery was utilized for further analysis. To simplify imagery interpretation, we clipped all imagery to only include a 200 m buffer around the Iowa DNR's stream polygon layer [41] (Figure 4).

Before use, the clipped images were visually assessed to ensure that they captured all recent stream channel changes that deviated from the stream polygon lines and moved beyond the 200 m buffer field of view.

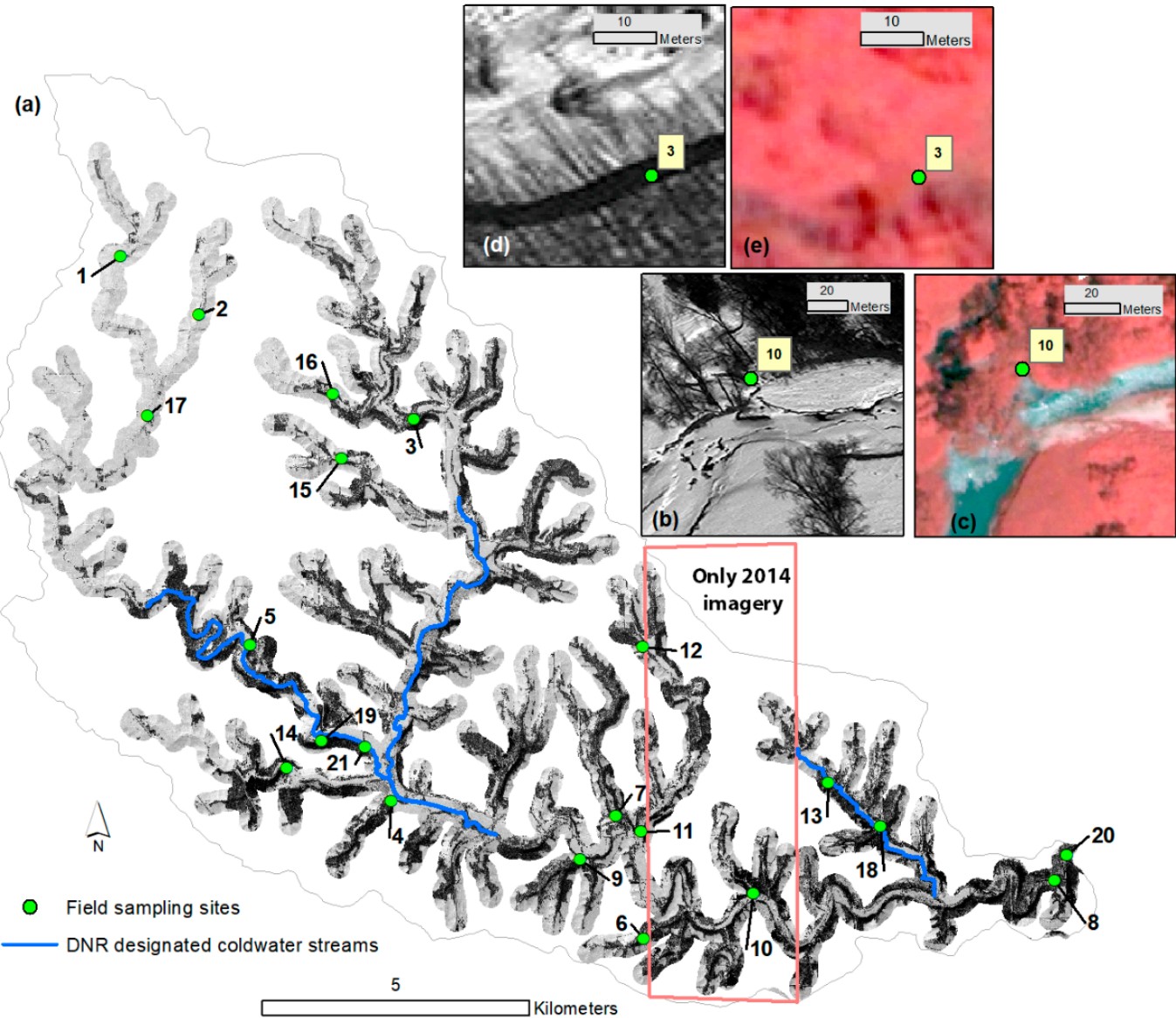

**Figure 3.** (**a**) Panchromatic imagery cropped to the 200 m buffer generated using the stream polygon vector (shown in Figure 1); the red polygon indicates the area where imagery from only 2014 was available; fish community survey locations are shown as green dots. (**b**,**c**) Comparison of winter panchromatic imagery and summer false color multispectral imagery for Site 10 (**d**) and Site 3 (**e**). Site IDs (1–21) of each fish community surveys are also shown.

To assess the influence of groundwater on stream segments, panchromatic imagery was visually interpreted by three trained analysts that identified stretches of open water that suggested a high probability of being coldwater. Imagery from 2014 and 2016 was compared, revealing spatio-temporal variability in stream conditions, making interpretation challenging. For example, stream reaches that were ice-free for one winter period sometimes appeared completely ice-covered in another image year. Differences were likely a result of variation in stream flows and/or air temperatures prior to satellite image acquisition. Topography, stream size, stream incision, and vegetation along the stream corridor also impacted analyst's ability to decisively interpret the imagery. Finally, image quality (e.g.,

low sun angles resulting in shadows) also complicated imagery interpretation. Because of these challenges, a two-tiered rating system was developed to assist analysts in designating coldwater reaches.

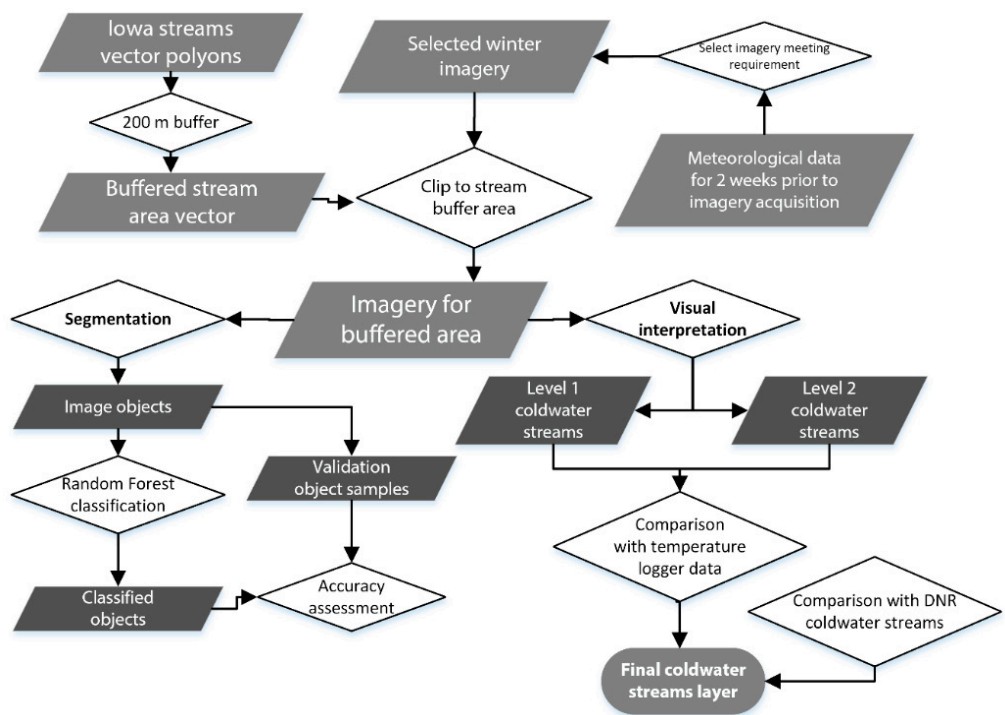

**Figure 4.** Illustration of the workflow used to develop and interpret the winter imagery for locating coldwater streams in the Canoe Creek watershed.

Stream reaches assigned a Level 1 rating were interpreted as open water by all analysts in both panchromatic images and often represented optimal conditions for coldwater identification (e.g., little to no canopy cover, high sun angle with minimal shadows). Stream reaches assigned a Level 2 rating were only identified as coldwater in one image or were collected with suboptimal site conditions (e.g., presence of canopy, low sun angle, and extensive shadows). In all cases, at least two of the analysts had to agree that the stream was ice-free for the reach to be classified as Level 2 coldwater.

To expedite image processing, we also tested semi-automatic image classification for detecting coldwater stream reaches. The panchromatic images were analyzed using geographic object-based image analysis (GEOBIA) in eCognition software (version 9.3) to map water as a proxy of coldwater streams. The GEOBIA approach first segments the imagery into objects representing ecologically meaningful patches on the landscape. Segmentation is followed by classification of objects into classes of interest using various rules and methods [47]. To distinguish water from other classes of interest (i.e., snow, vegetation/shade) we enhanced the edge or sharp contrast in the imagery by applying the edge extraction Lee Sigma filter to create a bright and dark edge layer. Later, the dark edge layer was added to and the bright edge layer was subtracted from an inverted version of the panchromatic band. This method enhanced the edge contrast along the water areas.

The resultant edge-enhanced modified panchromatic band was segmented using a multi-resolution algorithm to generate objects by merging several pixels together based on relative homogeneity criteria [48]. These criteria were defined by setting values for the scale, shape/color, and compactness/smoothness parameters. The optimal values for these parameters were determined by visually comparing the segmentation results of multiple iterations for different parts of the study area. Final classification was conducted using an ensemble decision tree classifier called random trees [49]. Representative object samples were selected for each class, and 2/3 of the samples were used for training the

classifier while 1/3 were used for accuracy assessment. Eighteen candidate features were selected for classification and involved spectral, geometry, and shape features. The best features for classification from these candidate features were selected using the feature space optimization (FSO) available within eCognition. Post-classification mapping accuracy was assessed by visually examining the results and using independent sample objects of each class. Finally, the results of visual interpretation and semi-automated GEOBIA analysis were spatially compared with the existing IA DNR coldwater stream layer to assess spatial agreement.

### 3. Results

#### 3.1. Water Temperature Monitoring

All but two of the locations monitored met the Iowa thermal criteria for coldwater stream designation (i.e., maximum summer temperature $\leq$ 75 degrees; Table 1). Temperature logger-obtained in situ data shows that (i) upstream reaches of open water that were closer to the mouth of springs showed consistently low temperature and variability, (ii) downstream reaches of coldwater streams showed more variability in temperatures but were still open water in winter imagery, and (iii) warmwater stream reaches exceeded the temperature threshold criteria (<75 degrees) and appeared frozen in winter imagery (Figure S1). Furthermore, out of the 21 water temperature-confirmed coldwater sites, 19 were identified correctly in image interpretation and 6 were interpreted as level 2 coldwater streams using satellite image interpretation (Table 1). This could be attributed to either canopy or terrain shading or bank shading creating hindrance in stream identification, since 67% of Level 2 sites had some amount of imagery obstruction.

**Table 1.** Water temperature monitoring results obtained from temperature loggers for selected locations (shown in Figure 2) and comparison with satellite imagery-based assessment.

| Site ID | Stream Width (m) from 2016 Imagery | Temperatures Support Coldwater Status? (Years that Support Classification) | Imagery Ice-Free? | Canopy or Terrain Shade Obstruction? | Image Interpretation Correctly Predicted Coldwater Status? |
|---|---|---|---|---|---|
| A | 13.7 | YES | YES (Level 1) | NO | YES |
| B | 4.3 | YES | YES (Level 1) | NO | YES |
| C | 3.4 | YES | YES (Level 1) | NO | YES |
| D | 1.5 | YES | YES (Level 1) | NO | YES |
| E | 3.9 | YES | YES (Level 1) | NO | YES |
| F | 2 | YES | YES (Level 1) | NO | YES |
| G | 2.6 | YES | YES (Level 2) | YES | YES |
| H | 3 | YES | YES (Level 2) | NO | YES |
| I | 4.2 | YES | YES (Level 1) | NO | YES |
| J | 12.5 | YES (1 out of 2) | YES (Level 1) | NO | YES |
| K | 8.5 | YES | YES (Level 2) | NO | YES |
| L | 2.9 | YES | YES (Level 1) | NO | YES |
| M | 8.7 | NO (1 out of 6) | NO | NO | YES |
| N | 3.1 | YES | YES (Level 2) | YES | YES |
| O | 2.5 | YES | YES (Level 1) | YES | YES |
| P | 2.6 | YES | YES (Level 1) | NO | YES |
| Q | 2.9 | YES | YES (Level 1) | NO | YES |
| R | 7.5 | NO (2 out of 8) | NO | NO | YES |
| S | 6.4 | YES (1 out of 1) | NO | NO | NO |
| T | 3.8 | YES (1 out of 1) | YES (Level 1) | NO | YES |
| U | 4 | YES (1 out of 1) | YES (Level 2) | YES (some bank shading) | YES |
| V | 4.1 | YES (2 out of 2) | YES (Level 2) | YES (some bank shading) | YES |
| W | 2.7 | YES (2 out of 2) | NO | YES (some bank shading) | NO |

### 3.2. Fish Communities

Of the 21 sites evaluated for the presence of coldwater fishes, trout were sampled at 18 sites (Table 2). Of the remaining sites, Site 6 contained no fish but went dry downstream of our survey site and had numerous bedrock barriers that could have inhibited fish immigration. Site 12 only had one species of fish (brook stickleback, *Culaea inconstans*), had the narrowest average wetted stream width of the surveyed sites, and was located very high up in the watershed. This site did, however, contain watercress, although no coldwater fishes were sampled. Finally, Site 9 did not have adequate water depth to complete a fish survey.

**Table 2.** Fish community survey results for selected locations and their comparison with satellite imagery-based assessment of coldwater habitat for those sites. The locations of these surveyed sites are shown in Figure 2.

| Site ID | Field-Derived Stream Width (m) | Fishery Supports Coldwater Status? | Imagery Ice-Free? | Canopy or Terrain Shade Obstruction? | Image Interpretation Correctly Predicted Coldwater Status? |
|---|---|---|---|---|---|
| 1 | 2.2 | YES | YES (Level 1) | NO | YES |
| 2 | 1.9 | YES | YES (Level 2) | YES (some bank shading) | YES |
| 3 | 4.9 | YES | YES (Level 1) | YES (some bank shading) | YES |
| 4 | 2.3 | YES | YES (Level 2) | YES | NO |
| 5 | 1.5 | YES | YES (Level 2) | YES | YES |
| 6 | 2.3 | NO | NO | YES | YES |
| 7 | 5.5 | YES | YES (Level 1) | NO | YES |
| 8 | 5.6 | YES | YES (Level 1) | NO | YES |
| 9 | - | NO (location dry) | NO | YES | YES |
| 10 | 2 | YES | YES (Level 2) | YES | YES |
| 11 | 3.4 | YES | YES (Level 1) | NO | YES |
| 12 | 1.1 | YES (watercress observed) | NO | YES | NO |
| 13 | 2.7 | YES | YES (Level 1) | NO | YES |
| 14 | 2.9 | YES | YES (Level 1) | NO | YES |
| 15 | 4 | YES | YES (Level 1) | NO | YES |
| 16 | 4.1 | YES | YES (Level 2) | YES | YES |
| 17 | 2.7 | NO (1 brown trout sampled) | NO | NO | YES |
| 18 | 4.3 | YES | YES (Level 1) | NO | YES |
| 19 | 7.1 | YES | NO | YES | NO |
| 20 | 13.7 | YES | YES (Level 1) | NO | YES |
| 21 | 6 | YES | YES (Level 1) | NO | YES |

Satellite imagery-based interpretation of surveyed fish community sites to determine coldwater stream status largely agreed with field observations. For three of the surveyed sites (sites 6, 9, 12), open water status could not be determined because of canopy shadow or terrain obstruction (Table 2). As a result, these sites were listed as ice-covered since analysts could not unanimously agree they were ice-free.

### 3.3. UAS Imagery Assessment

The SfM processing of UAS images produced an orthomosaic (spatial resolution 0.03 m) for each of the four sites and represented summer conditions. Visual interpretation and comparison of the UAS-derived orthomosaic with satellite-acquired VHR winter imagery allowed identification of very fine scale spatial patterns in the spatial distribution of coldwater streams. For example, at Site 1, the UAS orthomosaic was able to clearly map coldwater stream stretches that were only 2 m wide and were not identifiable in the satellite-acquired winter VHR imagery (Figure 5c).

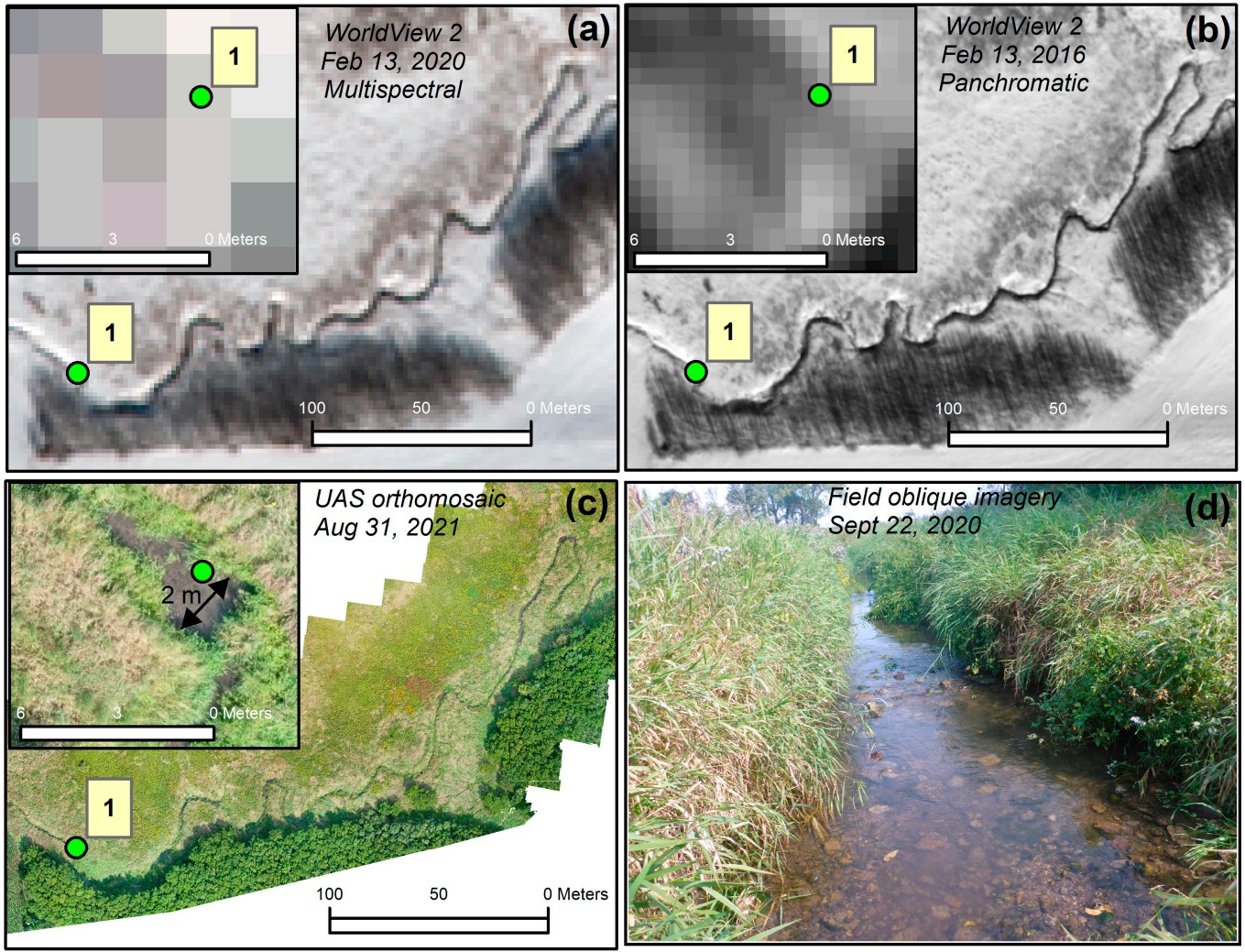

**Figure 5.** For Site 1: (**a**) multispectral imagery (2 m pixel size), (**b**) panchromatic imagery (0.5 m pixel size), (**c**) UAS-acquired imagery (0.03 m pixel size), and (**d**) oblique field photo.

### 3.4. VHR Winter Imagery Assessment

To semi-automatically detect coldwater stretches, we applied GEOBIA classification to the panchromatic imagery. Figure 6 shows the results for two different stream scenarios. Figure 6a shows an upper reach of the watershed with an average channel width of 3–5 m and no significant canopy influence. The GEOBIA approach detected open water, but it also frequently misclassified tree trunks, branches, and shadows of vegetation as water (Figure 6a) while parts of the stream channel were also incorrectly identified as vegetation. Drifting snow across the stream channel also made coldwater stream identification difficult in certain situations by obscuring the stream and limiting the detection of coldwater based on remotely sensed imagery. The GEOBIA approach was found to be more accurate in wider stream reaches when the stream width was >10 m. Figure 6b shows the results for a downstream reach with an average width of 15 m. While there were a few misclassifications (e.g., canopy shadow or trees misclassified as water and water misclassified as vegetation), overall the GEOBIA method produced acceptable results. However, most coldwater stream reaches in northeast Iowa are narrow and do not fit into this category, making correct identification with the GEOBIA approach difficult. When applied to entire-watershed scale mapping, the GEOBIA approach took significant processing time and produced low accuracies. As a result, semi-automatic GEOBIA classification was not used for this assessment.

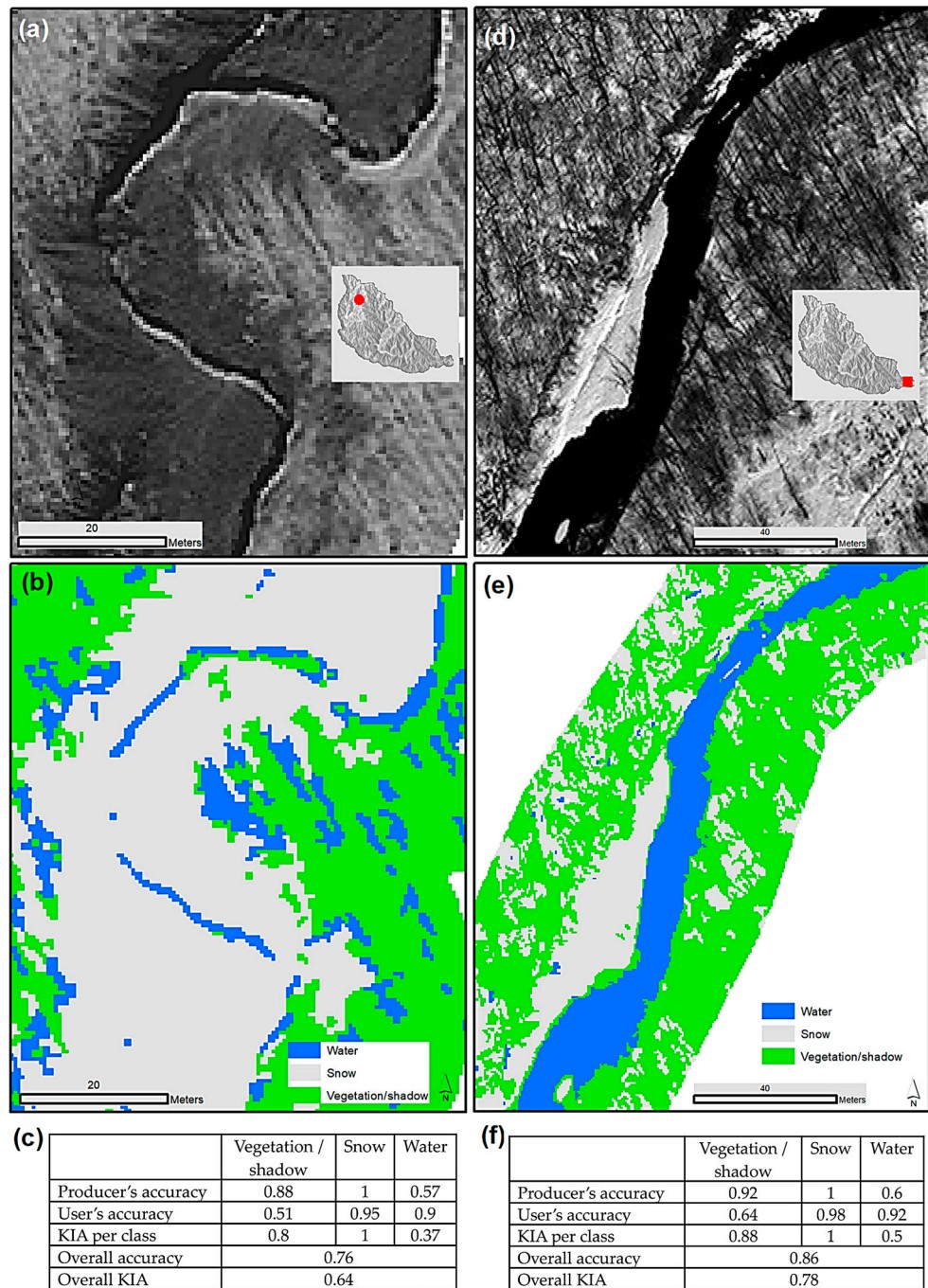

**Figure 6.** Results of GEOBIA for two selected stream reaches: (**a**) shows an upstream reach with a very narrow stream, (**b**) GEOBIA classification result and (**c**) classification accuracy for the upstream reach, (**d**) represents a much wider downstream reach, (**e**) shows the GEOBIA classification result and (**f**) shows the classification accuracy for this area.

Visual assessment of winter imagery in the Canoe Creek watershed detected a total of 57.3 km of ice-free stream segments, suggesting the presence of coldwater stream conditions. Of the identified coldwater segments, 35.4 km was interpreted as Level 1 while 21.9 km was classified as Level 2 (Figure 7a). Although 23.1 km of our mapped coldwater stream reaches is currently classified as coldwater by the Iowa DNR (Figure 7b), our visual assessment also detected 34.2 additional km of potential coldwater stream habitat that is currently not classified as coldwater (Figure 7c). There was also 3.2 km of stream currently classified as coldwater that was not identified in our winter imagery analysis (Figure 7d).

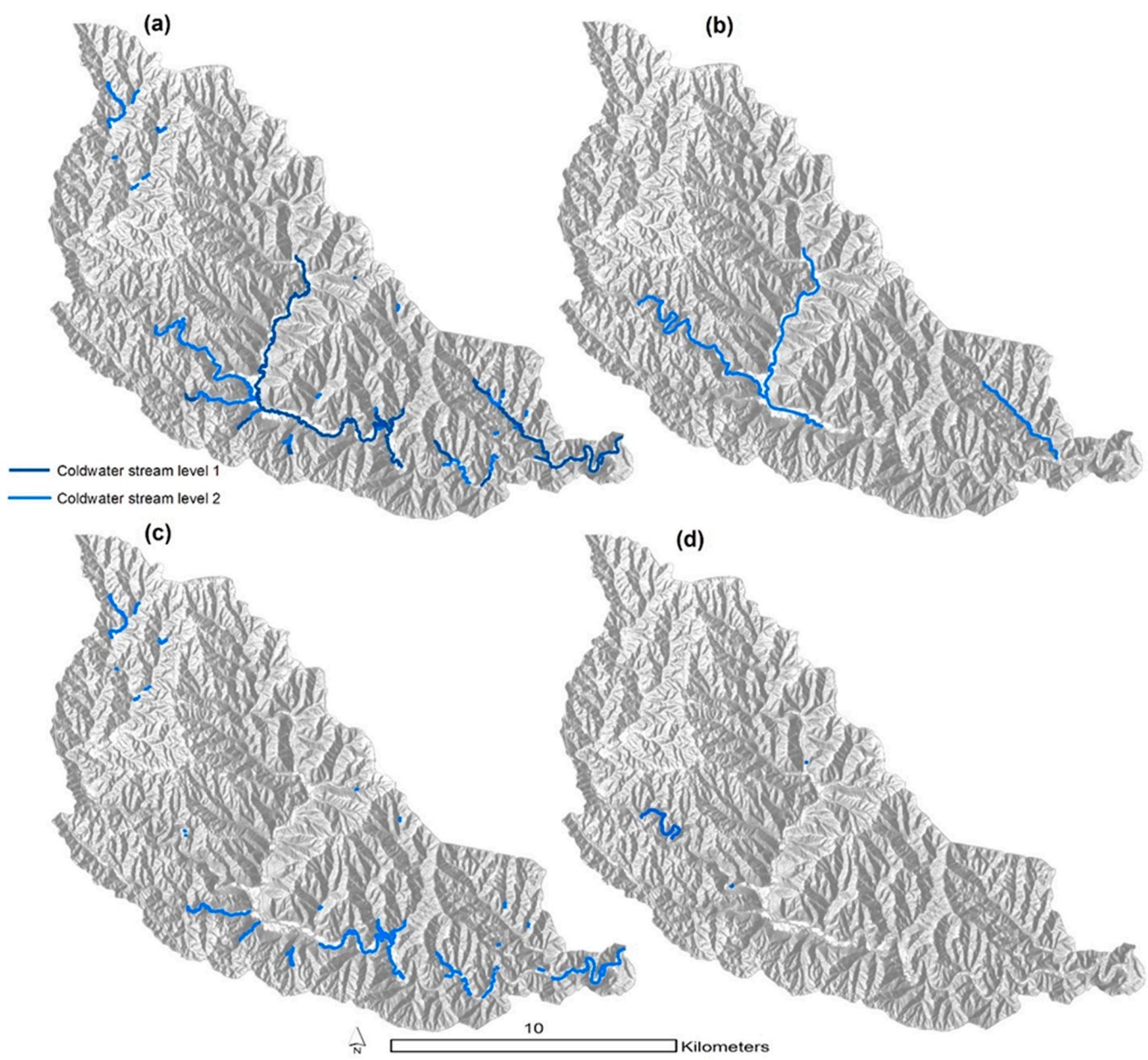

**Figure 7.** (**a**) Coldwater stream reaches identified by visual interpretation of VHR winter imagery combined with fish community and stream temperature data in this study; (**b**) coldwater stream reaches currently classified by IA-DNR as coldwater; (**c**) stream reaches that are not currently classified as coldwater but were identified as coldwater in our analysis; and (**d**) stream reaches currently classified as coldwater by IA-DNR but which were not identified as coldwater in our analysis.

## 4. Discussion

Using VHR winter images, this study was able to accurately detect where high-quality coldwater habitat existed in the northeast Iowa landscape. In addition to the location, we were also able to determine the general extent of each coldwater reach. Using our approach, we were able to confirm the current coldwater designation of 23 km of stream while also discovering an additional 34 km of stream that likely meets the evaluation criteria for reclassification from warmwater to coldwater surface water status. At nearly every location where we validated our imagery assessment using field data, the data supported the imagery-based recommendation (Tables 1 and 2). In some cases, the field data did not support the imagery-based recommendation; however, in all those cases, the imagery-

based recommendations were more conservative than those based on water temperature or fish community field data. For example, at Site 19, the fish community data supported coldwater status although the segment was ice-covered at that location. In another case, limited water temperature data at sites S and W supported coldwater status although the stream segment was ice-covered in the imagery.

Major factors can influence the effectiveness of coldwater stream identification from winter satellite imagery including the spatial resolution of the image, stream width, time of image acquisition, image acquisition angle, presence of canopy cover, and local topography. For stream reaches >10 m in width, both panchromatic and multispectral imagery are suitable for coldwater stream identification. As stream widths narrow, particularly below 5 m, multispectral imagery (with pixel resolution >1 m) is suboptimal for open water or coldwater stream determination. When using winter imagery to locate ice-free reaches, panchromatic imagery, with its higher spatial resolution (Figure 8b), was better suited than multispectral imagery, since many coldwater stream reaches we examined were <5 m in width (Tables 1 and 2; Figure 8d). Coldwater stream reaches that are 2 to 3 m wide are even sometimes difficult to detect with panchromatic imagery and require higher spatial resolution (e.g., UAS) imagery (Figure 8c). Therefore, UAS imagery has the potential to be utilized for validation of satellite imagery-derived mapping results. Besides spatial resolution, the timing of image acquisition is also an important consideration. Imagery captured in the spring, when deciduous tree canopy is minimal, was better for visualizing streams compared to imagery captured in the summer due to restricted views of the streams from increased canopy cover (e.g., Figure 4).

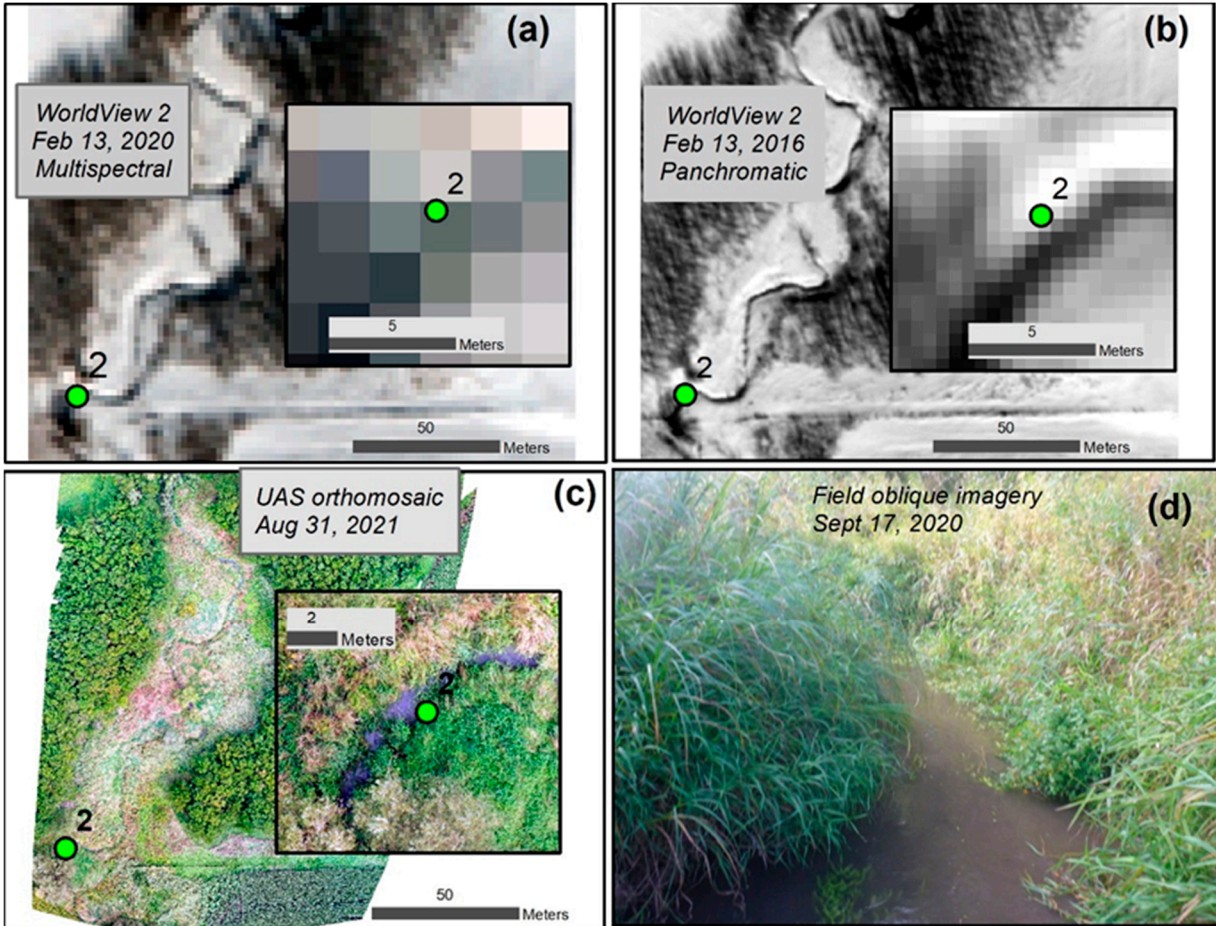

**Figure 8.** (**a**) Multispectral imagery (2 m pixel), (**b**) panchromatic imagery (0.5 m pixel), (**c**) UAS-acquired imagery (0.03 m pixel size), and (**d**) oblique field photo for Site 2.

Coldwater stream detection using VHR winter imagery in areas with significant canopy coverage is much more challenging than in areas lacking canopy and may have limited potential. For example, at Site 4, trout were successfully sampled during fish surveys, confirming it was a coldwater reach (Table 2). While thick canopy cover in the summer imagery hindered stream detection (Figure 9c), the oblique image acquisition angle and shadows of tree trunks and branches in winter imagery for both years (Figure 9a,b) significantly limited its usefulness. Therefore, this stream reach could only be assigned a Level 2 coldwater rating. For locations such as Site 4 where winter imagery has limited potential, field photos (Figure 9d) and biological sampling data may be necessary to determine the stream thermal designation. Even with the lack of foliage during the capture of winter imagery, shaded portions of the stream created by tree trunks and branches made the identification of coldwater stream reaches challenging. Unlike remotely sensed imagery that is collected from nadir geometry (e.g., Landsat), most of the VHR imagery is acquired from an oblique (off-nadir) view angle which is suboptimal for image interpretation. Furthermore, at higher latitudes in winter (in the northern hemisphere), the low sun azimuth angle increases the impact of canopy and terrain, resulting in significant shading which makes visual interpretation harder.

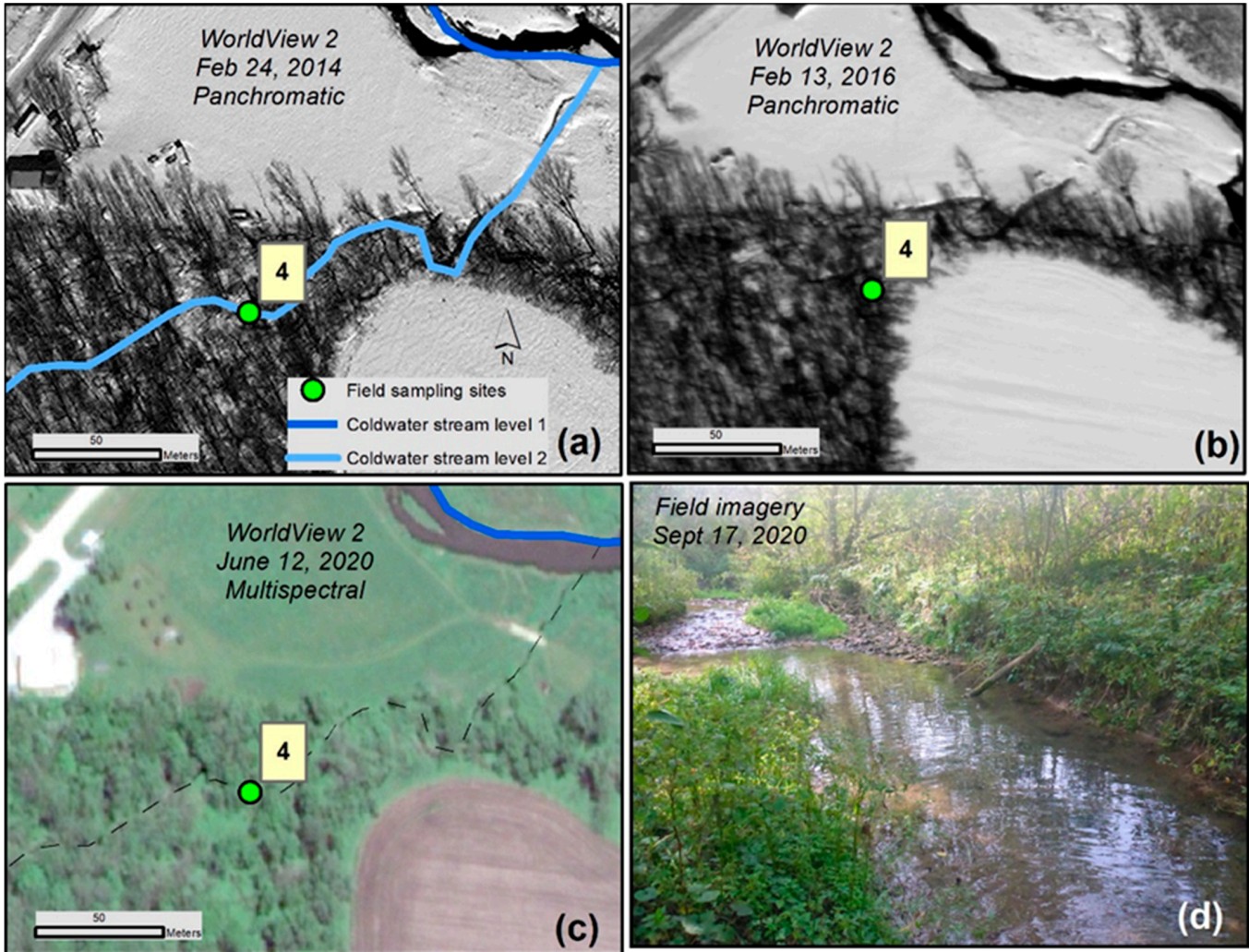

**Figure 9.** (**a**,**b**) Dense canopy cover limiting coldwater stream identification from winter imagery; (**c**) summer imagery limiting stream detection due to dense overhead canopy; and (**d**) oblique field photo of Site 4 where trout were sampled, supporting classification as a coldwater stream.

This study demonstrated the potential and limitations of using winter VHR imagery for identifying coldwater streams at watershed scales. While one previous study [16] examined this potential for a limited number of sites, ours is the first study to extensively test the potential of this approach for watershed scale mapping using an extensive network of critical field-acquired temperature logger and fish community survey data for validation. Validating remotely sensed imagery-derived results against field data enabled us to show the potential challenges and pitfalls of our ability to identify coldwater streams based on VHR imagery alone. Our results show that the spatial extent of coldwater habitat is temporally highly variable. Changes in detected coldwater extent over years can be attributed to complex site conditions and winter image quality and availability that determined the limits of detection. Our results underscore that although VHR winter satellite imagery has potential to be used for delineating coldwater inputs over larger geographical extents, knowledge of physiographic conditions and hydrological processes in the study area is essential to avoid misclassifications.

The influence on headwater streams of groundwater inputs is highly dependent on aquifer geometry and composition [50], location of the groundwater source relative to the stream [51], and seasonal temperature and precipitation patterns [38]. Recently, the groundwater discharge rate and baseflows in the streams of the Driftless Region has changed due to increases in precipitation throughout the year as well as land use changes [52,53]. Besides long-term trends in groundwater flows, annual variation in flow rates can also influence ice coverage rates observed in winter imagery in a given year. Long-term trends and short-term annual changes to baseflows could significantly influence the spatial extent of coldwater streams over time. The methodology to detect coldwater streams using VHR winter imagery, as shown in this study, could be used to detect these temporal changes in the extent of coldwater habitat.

It is also important to consider that there is some amount of uncertainty in the derived coldwater stream length, since any magnitude of uncertainty in the original stream length layer is transferred to the final coldwater stream dataset. This study utilized the IA DNR's stream polygon layer for Iowa. Sometimes these layers are not developed using the highest resolution base imagery available and they are therefore prone to underestimating the actual stream length, especially in headwater areas where the stream width is narrow and significant canopy cover exists [54]. In some cases, small coldwater spring runs or seeps may not even be designated as a stream segment. Furthermore, spatial comparison of our results to the segments classified by the IA DNR as coldwater showed substantial spatial mismatch (Figure 10a). It is important to consider that stream channels are dynamic and can spatially shift or laterally migrate over time in response to changing discharge related to floods and other extreme weather events. In spite of this spatial disagreement in certain areas, we observed close spatial agreement in the stream channel pattern for the majority of the study area (>90%).

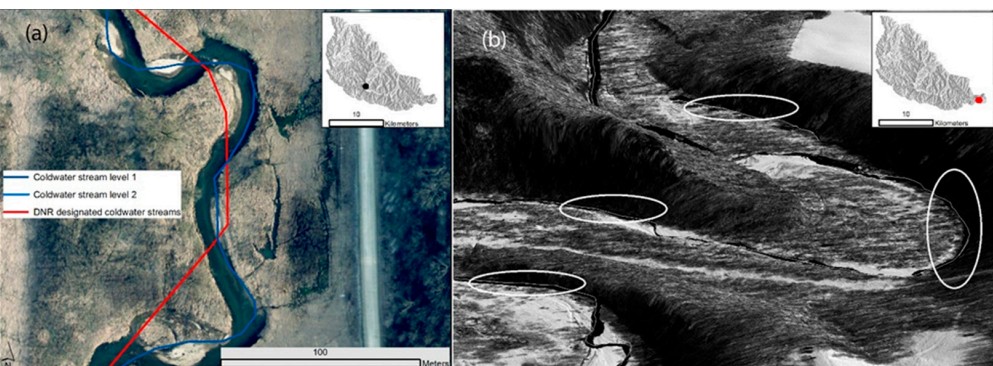

**Figure 10.** (**a**) Example of spatial mismatch/disagreement between the DNR's and this study's interpretations of a coldwater stream and (**b**) terrain shadow impacting coldwater stream visualization/interpretation in winter panchromatic imagery.

VHR imagery-based coldwater stream detection has value as long as good-quality winter imagery is available. Since the early 2000s, the number of operational commercial VHR satellite sensors has increased, and the VHR optical imagery archive is rapidly expanding. However, in our search of the Digital Globe /MAXAR VHR optical imagery archive across a 10-year time window (2010–2020), we found that >95% of the available imagery for our study area was acquired during summer and was unsuitable for coldwater stream identification. Recent studies conducted in northern temperate climates [16] have found potential in publicly available winter imagery in Google Earth Pro. However, image availability can be spatially variable, as in our experience. For the Canoe Creek watershed, we did not find any winter imagery in Google Earth Pro for the last 25 years.

## 5. Conclusions

Identifying coldwater stream reaches is critical so that these unique and valuable resources can be protected and properly managed. Coldwater identification using VHR winter imagery exclusively can be difficult due to complex stream conditions and site variables. Our recommendations for imagery-based coldwater classification were more conservative than what would have been recommended at the site when also considering field data, suggesting that imagery generally detects only the highest quality coldwater habitat. Lower quality stream segments that still meet the thermal or biological criteria to be classified as coldwater could remain undetected if using only the imagery-based approach. Field obtained temperature, habitat, and fish survey data provide accurate representations of selected point locations and are critical to overcome these challenges. UAS imagery can provide the necessary spatial resolution to characterize coldwater stream habitat on narrow streams and should be used to bridge the gap between in situ-obtained data and satellite imagery. Regardless, imagery-based detection is a quick and relatively inexpensive method to rapidly locate coldwater resources in areas with limited field data, fill data gaps, and add to existing coldwater datasets. Our results will be used to recommend updates to the thermal designation of stream reaches within the Canoe Creek watershed. Ongoing research is focused on expanding this methodology to detect coldwater segments for the entire Driftless Region of Iowa. Future research could use thermal infrared imagery acquired using a UAS platform to better characterize the spatio-temporal variability in stream temperatures and compare them across multiple coldwater habitat sites.

**Supplementary Materials:** The following supporting information can be downloaded at: https://www.mdpi.com/article/10.3390/rs15184445/s1, Figure S1: In situ obtained temperature profile for three selected locations. The locations K and P meet the State of Iowa coldwater stream designation criteria (maximum summer temperature <75 degrees) while location R fails to meet these criteria.

**Author Contributions:** Conceptualization, N.B.M. and M.J.S.; methodology, N.B.M. and M.J.S.; software, N.B.M.; validation, N.B.M., M.J.S. and G.S.; formal analysis, N.B.M.; writing—original draft preparation, N.B.M.; writing—review and editing, N.B.M., M.J.S. and G.S.; visualization, N.B.M.; supervision, N.B.M.; project administration, N.B.M.; funding acquisition, N.B.M. and M.J.S. All authors have read and agreed to the published version of the manuscript.

**Funding:** This research was funded by the DOI Fish and Wildlife Management Assistance Program (grant #F20AP10272-00).

**Data Availability Statement:** Upon publication, the data and results will be available via the first authors' GitHub platform.

**Acknowledgments:** The University of Iowa State Hygienics Lab and the Weber Fisheries Ecology and Management Lab at Iowa State University provided fish community and water temperature data. The authors thank Ethan Wedemayer, Grace Kunkel, and Ali Chalberg for help with visual interpretation; Louise Mauldin and Jeena Koenig for assistance with high resolution imagery search; Eric Strauss for guidance and helpful discussions; and Iowa DNR personnel for assisting in UAS imagery collection. We also thank anonymous referees for their constructive and insightful comments to help improve this work.

**Conflicts of Interest:** The authors declare no conflict of interest. The funders had no role in the design of the study; in the collection, analyses, or interpretation of data; in the writing of the manuscript; or in the decision to publish the results.

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
