# Peer review of "Applying High-Resolution Satellite and UAS Imagery for Detecting Coldwater Inputs in Temperate Streams of the Iowa Driftless Region"

_remotesensing, doi:10.3390/rs15184445_

Round 1
Reviewer 1 Report
Summary Comment – The topic of the paper is novel and interesting. Mapping coldwater streams is a very challenging remote sensing application. In addition, the paper is generally well-written and well-organized, however, although the authors explored lots of remote sensing approaches, they didn’t end up presenting much quantitative results for anything other than the visual interpretation effort, which is problematic given the journal focus. Please see the specific comments below.
Major Comments
Lack of remote sensing results – although the effort creatively integrates multiple types of data to answer a challenging question, the remote sensing piece, unfortunately, really isn’t there. For example the authors collect UAS imagery, but other than provide a Figure (Figure 5), there doesn’t seem to be any analysis performed on it. The authors also thoroughly describe the image selection and GEOBIA process for 2 panchromatic images in the Methods, but then state in the results that the GEOBIA was not used for this classification and the results seems to be qualitative interpretations. This leaves the only remote sensing included to be the visual interpretation of the 2 panchromatic images, which really isn’t a great fit for a Remote Sensing journal. I recommend adding in an analysis and results for both the UAS and the GEOBIA effort. For example, for the GEOBIA, report the random forest classification model accuracy, and add the GEOBIA classification results to Table 1 and 2. You can then use the less than ideal accuracy to justify selecting the visual interpretation data. For the UAS, can you quantify the value-add of the fine-scale spatial patterns somehow? Or does the UAS collection overlap with sites in Table 1 and 2 so that you could add an interpretation from that data to those tables?
Lines 267-268 – Please provide more details regarding the classification, how many objects was the classification trained on? What variables were included in the classification? Was it just mean object panchromatic value or were others used? How many trees? What other model parameter values were selected? Was it a binary classification or were there multiple classes?
Methods – there were a lot of different datasets to keep track of. Please add a table that summarizes all of the input data, the images, UAS collection, water temperature monitoring data and fish surveys, with the number of images/sites, dates of collection, etc.
Minor Comments
Line 128 – add example temperatures or temperature ranges here to support interpretation
Lines 144-149 – This is great text but it really doesn’t fit in the Methods, I would recommend moving this text to the end of the Introduction as it provides is a good justification for the effort.
Lines 233-240 – the use of the term “readers” is odd here, as it could be confused with readers of the article. What about using a more common term like “analyst” here?
Line 270 – what are the classes? Is this water/non-water or were other classes trained on as well? Please clarify in the text.
Section 3.1 and 3.2 – add a sentence to both sections summarizing the accuracy of the image interpretation…X of Y sites were predicted correctly…
Table 1 and 2 – is the “image correctly predicted coldwater status?” – I believe this is referring to the visual interpretation effort? Please clarify in the captions.
Line 315 – how can you tell they are coldwater stream reaches?
Line 333-334 – is this just a general observation? Could this be quantified?
Line 341 – what was the accuracy?
Lines 471-480 – What do you think you would need, r.s. data-wise to be able to accurately classify these and scale the approach? Would more winter data do it? Or 0.5 m multi-spectral data? What if you pan-sharpened multi-spectral data down to 0.5 m?
Author Response
Summary Comment – The topic of the paper is novel and interesting. Mapping coldwater streams is a very challenging remote sensing application. In addition, the paper is generally well-written and well-organized, however, although the authors explored lots of remote sensing approaches, they didn’t end up presenting much quantitative results for anything other than the visual interpretation effort, which is problematic given the journal focus. Please see the specific comments below.
Response: We thank the reviewer # 1 for their time in reviewing our manuscript and providing feedback to help improve the quality and scientific rigor of our research work. Below please find our response to your comments:
Lack of remote sensing results – although the effort creatively integrates multiple types of data to answer a challenging question, the remote sensing piece, unfortunately, really isn’t there. For example the authors collect UAS imagery, but other than provide a Figure (Figure 5), there doesn’t seem to be any analysis performed on it. The authors also thoroughly describe the image selection and GEOBIA process for 2 panchromatic images in the Methods, but then state in the results that the GEOBIA was not used for this classification and the results seems to be qualitative interpretations. This leaves the only remote sensing included to be the visual interpretation of the 2 panchromatic images, which really isn’t a great fit for a Remote Sensing journal. I recommend adding in an analysis and results for both the UAS and the GEOBIA effort. For example, for the GEOBIA, report the random forest classification model accuracy, and add the GEOBIA classification results to Table 1 and 2. You can then use the less than ideal accuracy to justify selecting the visual interpretation data. For the UAS, can you quantify the value-add of the fine-scale spatial patterns somehow? Or does the UAS collection overlap with sites in Table 1 and 2 so that you could add an interpretation from that data to those tables?
Response: While relevant methodological details about image selection, pre-processing and object based image analysis are presented the quantitative results from the OBIA/RF classification were not discussed in detail because this method was unable to detect coldwater streams with acceptable/usable accuracy for the majority of the study area. From the Initial analysis of field obtained stream habitat conditions and fish community data and its visual comparison with VHR imagery for selected locations it was evident that many of the coldwater stretches were challenging to even visually identify (due to various site and environmental conditions and as shown in shown in Figure 5 and 8). Significant portions of the study area exhibited these conditions and even before the automated classification, we had low expectations of accurately detecting such areas with the OBIA/RF method. During the analysis we did perform OBIA/RF classifications for several iterations by optimizing the feature space by including various object features (e.g. spectral, geometric, shape) and by iteratively refining the training and validation object sample quality. We presented only the most necessary details of image selection and OBIA/RF analysis and skip the details of iterative OBIA/RF trails as we did not find much value in adding them in the manuscript as none of them improved the classification accuracy to acceptable levels. Nonetheless, we presented the classification results for two selected areas to illustrate where the OBIA/RF methodology worked and where it did not. The revised manuscript now includes accuracy matrices (producers and users’ accuracy, kappa) for these two sites.
Rather than mapping coldwater stretches by selecting point locations and doing a “proof-of-concept” (as some previous studies have done e.g. O’Sullivan et al 2019), our objective was to be able to test winter VHR imagery for watershed scale detection of coldwater stretches which will allow usable implications for environmental managers. Knowing the fact that UAS imagery can not practically be used for watershed scale monitoring, we used UAS imagery for 4 selected locations solely for validating the results of VHR imagery. Although an OBIA/RF or similar analysis methods could have been applied to UAS imagery, such analysis would not meet the objectives and requirements of this study. Therefore, UAS orthomosaic was used to characterize stream and habitat conditions in detail and for validation of VHR imagery derived results.
In a situation where automated analysis failed to produce acceptable levels of accuracy, visual interpretation was deemed the suitable way forward by comparing the interpretations of several interpreters and by using UAS and ground obtained data. We believe by combining and comparing high quality field data with the analysis of multiscale imagery our work is able to present and comment on both remote sensing data and methodological challenges for coldwater streams mapping which very few previous studies have done. Our findings will be a useful reference for future studies and therefore a good fit for a remote sensing journal.
Lines 267-268 – Please provide more details regarding the classification, how many objects was the classification trained on? What variables were included in the classification? Was it just mean object panchromatic value or were others used? How many trees? What other model parameter values were selected? Was it a binary classification or were there multiple classes?
Response: The classification involved 3 classes (water, snow and vegetation/shadow) which are depicted in the classification figure legend (figure 6). Below please find the details of the OBIA/RF classification procedure requested by the reviewer. We believe including these details will not add value to the manuscript and rather make it unnecessarily long (particularly because the OBIA/RF obtained results were not used for further analysis in the manuscript):
Both Figure 6(b) and 6(e) show only about 10% of the actual area that was classified for each region. The total number of objects for classification shown in 7(b) was 28,523 objects while the total number of objects for classification shown in7(e) was 66,785 objects. Representative object samples were selected for each class and 2/3 of the samples were used for training the classifier while 1/3 were used for accuracy assessment. Eighteen candidate features were selected for classification and involved spectral, geometry and shape features. The best features for classification from these candidate features were selected using the Feature Space Optimization (FSO) available within eCognition. FSO evaluates the Euclidean distance in the feature space between the samples of all classes and selects a feature combination resulting in the best class separation distance, which is defined as the largest of the minimum distances between the least separable classes. The following ten features were selected by FSO and used for RF classification: Mean (EEMPB), Standard Deviation (EEMPB), Asymmetry (EEMPB), Border index, Compactness (EEMPB), Density (EEMPB), Elliptic fit (EEMPB), Rectangular fit (EEMPB), Roundness (EEMPB), Shape index (EEMPB); where EEMPB refers to the Edge Enhanced Modified Panchromatic Band. The definition (formula) of each of these feature’s calculation can be found in eCognition user guide.
Methods – there were a lot of different datasets to keep track of. Please add a table that summarizes all of the input data, the images, UAS collection, water temperature monitoring data and fish surveys, with the number of images/sites, dates of collection, etc.
Response: We believe that existing figure 4 (flowchart showing the data utilized and workflow for analysis) is intended to help the readers understand what sources of data were used and at what step of analysis they were utilized. The manuscript currently has 10 figures and 2 tables and including another table will further increase the length.
Line 128 – add example temperatures or temperature ranges here to support interpretation
Response: Please see Supplementary figure 1 (Figure S1) which shows in situ obtained temperature profile for three selected locations. The selected locations show the temperature ranges, two of which meets the State of Iowa coldwater stream designation criteria (maximum summer temperature < 75 degrees) and one that does not meet the criteria.
Lines 144-149 – This is great text but it really doesn’t fit in the Methods, I would recommend moving this text to the end of the Introduction as it provides is a good justification for the effort.
Response: Thank you. This was moved to the end of introduction section.
Lines 233-240 – the use of the term “readers” is odd here, as it could be confused with readers of the article. What about using a more common term like “analyst” here?
Response: The term readers was replaced by analyst. Thank you.
Line 270 – what are the classes? Is this water/non-water or were other classes trained on as well? Please clarify in the text.
Response: The three mapped classes were water, snow and vegetation/shadow. Results are shown in figure 6 which includes legend showing these classes.
Section 3.1 and 3.2 – add a sentence to both sections summarizing the accuracy of the image interpretation…X of Y sites were predicted correctly…
Response: We agree and have added suggested text.
Table 1 and 2 – is the “image correctly predicted coldwater status?” – I believe this is referring to the visual interpretation effort?
Response: Yes, this refers to our results from visual interpretation. We have added text to clarify this.
Line 315 – how can you tell they are coldwater stream reaches?
Response: We agree with the reviewer that during this season we would not be able to determine if it was a warmwater or coldwater based on UAS imagery alone, and we have updated our text to discuss how UAS is better suited for mapping narrow stream segments in general.
Line 333-334 – is this just a general observation? Could this be quantified?
Response: Accuracy matrices (producers, users and kappa) are included in the revised manuscript.
Line 341 – what was the accuracy?
Response: Accuracy matrices (producers, users and kappa) are included in the revised manuscript.
Lines 471-480 – What do you think you would need, r.s. data-wise to be able to accurately classify these and scale the approach? Would more winter data do it? Or 0.5 m multi-spectral data? What if you pan-sharpened multi-spectral data down to 0.5 m?
Response: Higher number of winter imagery will help. However as shown by our results, scaled up mapping based on classified VHR satellite imagery alone will not be able to detect most of the coldwater areas that are either too narrow in width (< 4 m) or are under terrain or canopy shadow. Unfortunately, there are too many coldwater streams throughout driftless Iowa that fall into this category. Field collected data on temperature, stream physiographic properties (width, vegetation cover and height along stream banks and vegetation canopy percentage) coupled with strategically collected UAS imagery (<0.1 m pixel resolution) for sites that are hard to reach will be required to determine the actual coldwater stream length over watershed scales. Pansharpening has potential to be helpful and can be done if multispectral is available with PAN imagery. In our case, pansharpening was not an option because PAN imagery was acquired in winter while the multispectral imagery was from summer. Pansharpening may change the range of pixel values compared to the original data (but this could be an issue for spectral modeling but not for image classification).

Reviewer 2 Report
Coldwater streams are an important resource in the context of global warming.This study demonstrated the potential and limitations of using winter VHR imagery for identifying coldwater streams at watershed scales. The identification of cold water flow in this paper has certain practical significance for its protection and management. On the whole, the article is well written. However, some problems in the article still need to be further modified to meet the requirements of publication.
Abstract
I think your abstract is too much and can be further streamlined to be less than 300 words.
Key words
(1)The hyphen in “surface water—ground water interactions” is not an English symbol, it should be “-”
(2)One of the role of keywords is clear and intuitive to express the theme of literature discussion or expression, so that readers can know the theme of the paper before reading the abstract and text. The number of keywords is generally 3-5, your article has 8 keywords, I think you should refine your keywords. In addition, the symbols between keywords should be unified. You used both commas and semicolons at the same time.
Introduction
What is the significance of your research? For example, what is the main question addressed by your research? At the same time, you should clarify the innovation of your research. How is it different from other research ?
Method
(1) I think Figure 1 should show elevation information, because elevation is also one of the important factors affecting water temperature and temperature.
(2) In remote sensing interpretation, you mentioned ' using a limited number of VHR winter images '. And why is the monitoring of water temperature only in spring, summer and autumn. Since the existence of groundwater makes some streams not frozen in winter, then I think the monitoring of these unfrozen streams is also interesting.
(3) Is there any objective standard for ' unusually warm and dry ' ? For example, how much is the temperature exceeding the anomaly, how do you judge the temperature anomaly.
(4) Where do these experienced readers come from? How to be experienced? For example, do they engage in related occupations that contribute to this remote sensing interpretation work ? Or do they have long-term experience in field practice? I think if there isn 't enough evidence to prove how experienced they are, then there are only three such people I think the results may be biased.
Results and discussion
(1) In the line 303-304 you wrote “these sites were listed as ice covered since viewers could not unanimously agree they were ice-free.” Viewers refer to these three experienced readers? Only because viewers do not all think that there is ice-free, it is a little random to list it as ice covered ?
(2) In the discussion, you discuss the use of VHR winter images to classify cold water flows, factors influence the effectiveness of cold water stream identification, the influence on headwater streams of groundwater inputs. I suggest that in the discussion section, it can be divided into 2-3 sub-titles according to the content, and further in-depth discussion.
(3) When mentioning factors influence the effectiveness of coldwater stream identification, you cite spatial resolution of image, image acquisition angle, time of image acquisition, stream width, presence of canopy cover. and local topography , but not in this order in the subsequent discussion, I suggest that the discussion of the influencing factors should be consistent with the order you say, so that the article is more organized.
Reference
Some references need further examination. References 26,30 lack page numbers. Reference 44 is incompleted, “(accessed on”?
Some charts in the article need further improvement.
(1)The text color of Fig 3 (d) should be consistent with the color of (a) (b) (c) (e). (d) (e) should also be bloded in the following illustration of Fig 3.
(2)The position of (a)(b)(c)(d) in Fig 5 should be consistent to ensure that the picture is more beautiful and tidy. And the frame of Fig 5 (c) is not shown in full
(3)The compasses in Fig.7 and Fig 6 are too small, and it should be placed in the upper right corner. And the style of the scale is also difficult to understand. It is recommended to change the scale of a style.
Minor editing of English language required
Author Response
Coldwater streams are an important resource in the context of global warming.This study demonstrated the potential and limitations of using winter VHR imagery for identifying coldwater streams at watershed scales. The identification of cold water flow in this paper has certain practical significance for its protection and management. On the whole, the article is well written. However, some problems in the article still need to be further modified to meet the requirements of publication.
Response: We thank the reviewer # 2 for their time in reviewing our manuscript and providing feedback to help improve the quality and scientific rigor of our research work. Below please find our response to your comments:
Abstract: I think your abstract is too much and can be further streamlined to be less than 300 words.
Response: Abstract was edited to make it more concise and streamlined.
Key words: (1)The hyphen in “surface water—ground water interactions” is not an English symbol, it should be “-”
Response: Corrected.
(2) One of the role of keywords is clear and intuitive to express the theme of literature discussion or expression, so that readers can know the theme of the paper before reading the abstract and text. The number of keywords is generally 3-5, your article has 8 keywords, I think you should refine your keywords. In addition, the symbols between keywords should be unified. You used both commas and semicolons at the same time.
Response: The number of key words were reduced, and the symbols were unified. Thank you for pointing out this inconsistency.
What is the significance of your research? For example, what is the main question addressed by your research? At the same time, you should clarify the innovation of your research. How is it different from other research?
Response: We believe that the abstract summarizes the research requirement/significance rather well. The main objective of this research is to show the potential and limitations of VHR satellite imagery for mapping coldwater streams in northern temperate climates. By combining extensive field data and UAS imagery with the analysis of VHR imagery our results show where VHR imagery-based detection works and where it fails. Further, citing specific site conditions and environmental and data related variables, we elaborate on the reasons of the VHR imagery based coldwater habitat detection. The discussion and conclusion sections summaries these finds and makes recommendations for future studies looking to utilize these datasets and methods.
Method: (1) I think Figure 1 should show elevation information, because elevation is also one of the important factors affecting water temperature and temperature.
Response: We agree that elevation could be an important factor affecting water and air temperature. We did not add information to Figure 1 as suggested by the reviewer; instead, we added text to the Study Area section describing the range of elevation encountered in the Canoe Creek watershed. We also included elevation information as part of Figure 2 by incorporating a hillshade background. Since elevation changes in the Canoe Creek watershed are relatively small, we believe this will address the concerns of the reviewer.
(2) In remote sensing interpretation, you mentioned ' using a limited number of VHR winter images '. And why is the monitoring of water temperature only in spring, summer and autumn. Since the existence of groundwater makes some streams not frozen in winter, then I think the monitoring of these unfrozen streams is also interesting.
Response: We agree that year-round monitoring of water temperatures is interesting and we have completed some limited work with year-round monitoring. In most cases, we install the water temperature monitors in the spring and remove them in the autumn to download data. In Iowa streams, the warm summer period is usually the limiting period for coldwater fishes and the primary reason streams are monitored during that period. For example, the Iowa Cold Water Use Designation Assessment Protocol states “The maximum stream water temperatures during mid-May through mid-September does not exceed 75°F under normal stream conditions…..”; therefore, we monitor stream temperatures during the required period.
(3) Is there any objective standard for ' unusually warm and dry ' ? For example, how much is the temperature exceeding the anomaly, how do you judge the temperature anomaly.
Response: We agree that the original text was unclear and we revised text to note that severe drought conditions resulted from “unusually warm and dry” weather.
(4) Where do these experienced readers come from? How to be experienced? For example, do they engage in related occupations that contribute to this remote sensing interpretation work ? Or do they have long-term experience in field practice? I think if there isn 't enough evidence to prove how experienced they are, then there are only three such people I think the results may be biased.
Response: We modified the text to better reflect the analysts. All three analysts were trained by the authors, reviewed and discussed the visual interpretation methods prior to the assessment of the study imagery. Training consisted of reviewing practice imagery and then discussing as a group how each analyst developed their recommendation. Interpretations were discussed as a group and a final recommendation developed. Once analysts were able to reliable located coldwater streams on practice imagery, they were allowed to begin working on the study imagery. Even with training, interpretations occasionally varied among analysts.
Results and discussion
(1) In the line 303-304 you wrote “these sites were listed as ice covered since viewers could not unanimously agree they were ice-free.” Viewers refer to these three experienced readers? Only because viewers do not all think that there is ice-free, it is a little random to list it as ice covered ?
Response: Yes, viewers refer to the three experienced interpreters. We have modified the text to refer to them as analysts as discussed above. Analysts have spent hundreds of hours looking at multi-year VHR satellite imagery from winter and summer seasons and have first had knowledge of the on-ground conditions (e.g. how terrain and shadow obstruction can significantly limit coldwater identification process) gained from fish community and habitat surveys in the Canoe watershed and throughout the driftless Iowa. The analysts were also trained to identify ice-free segments of streams prior to reviewing study imagery. Even with hours of experience and training, some stream segments are challenging to interpret; therefore, we made the judgement to not include these sections in coldwater streams (which reduces false positives).
(2) In the discussion, you discuss the use of VHR winter images to classify cold water flows, factors influence the effectiveness of coldwater stream identification, the influence on headwater streams of groundwater inputs. I suggest that in the discussion section, it can be divided into 2-3 sub-titles according to the content, and further in-depth discussion.
Response: We appreciate the reviewer’s input but believe that the several factors that influence coldwater stream detection occurs in conjunction rather than in isolated manner and discussion as written in current form (without sub-sections) allows as to show the whole story in an interconnected way. For example, both image properties (e.g. spatial resolution, solar elevation at which the imagery was acquired) and ground conditions (e.g. stream width, terrain or canopy shadow) impact coldwater stream detection ability. Generally higher spatial resolution of imagery allows better chances of coldwater stream detection. However, in terrain or canopy shadow areas, even if higher spatial imagery is available, it does not guarantee that we will be able to confidently detect coldwater stream presence or absence. Presenting the discussion without subsection allows us to cross-refer and cross-compare the influence of these multiple factors that create unique ground conditions that limit coldwater stream identification. Dividing the discussion into multiple sub sections will present the reader an isolated view and limit their understanding of the inter-connectedness of these data related and environmental factors.
(3) When mentioning factors influence the effectiveness of coldwater stream identification, you cite spatial resolution of image, image acquisition angle, time of image acquisition, stream width, presence of canopy cover. and local topography , but not in this order in the subsequent discussion, I suggest that the discussion of the influencing factors should be consistent with the order you say, so that the article is more organized.
Response: Thank you for pointing this out. The order in which the influencing factors were listed has been modified and now matches with the subsequent discussion.
Some references need further examination. References 26,30 lack page numbers. Reference 44 is incompleted, “(accessed on”?
Response: Thank you. These references are updated with page numbers and other required details.
Some charts in the article need further improvement.(1)The text color of Fig 3 (d) should be consistent with the color of (a) (b) (c) (e). (d) (e) should also be bloded in the following illustration of Fig 3.
Response: Thank you. In Fig 3, if the text figure in (d) is changed to black, it becomes indistinguishable due to its background being close to black. If the color of other labels is changed from black (to either gray or white) then they become indistinguishable from their background. Therefore, we did not change the text color for Fig 3 (d).
(2)The position of (a)(b)(c)(d) in Fig 5 should be consistent to ensure that the picture is more beautiful and tidy. And the frame of Fig 5 (c) is not shown in full
Response: Thank you. Fig 5 was revised to address these issues.
(3) The compasses in Fig.7 and Fig 6 are too small, and it should be placed in the upper right corner. And the style of the scale is also difficult to understand. It is recommended to change the scale of a style.
Response: Thank you. The placement and size of the north arrow and scale bar has been selected so as not to distract the focus of the viewer from the main thematic component shown in the figures.
Round 2
Reviewer 1 Report
This paper is a creative approach to applying high-resolution remote sensing to a challenging data gap. I appreciate the revisions the authors made, please address the 2 remaining comments below.
Lines 364-366 - I can appreciate that the authors are sensitive to the paper length, however, I firmly believe that the Methods need to be adequately described so that they could be replicated. In light of this, please add the Response to Comments text on the additional random forest details, in particular the variables included in the tree, but also the number of trees and any other parameters set, to the Methods section.
Figure 6 - I appreciate the authors adding confusion matrices to this figure to provide examples of how well the algorithm performs where it was found to work, however, details need to be added to the Methods text about how these validation points were collected and applied to derive these accuracy statistics.
Author Response
Lines 364-366 - I can appreciate that the authors are sensitive to the paper length, however, I firmly believe that the Methods need to be adequately described so that they could be replicated. In light of this, please add the Response to Comments text on the additional random forest details, in particular the variables included in the tree, but also the number of trees and any other parameters set, to the Methods section.
Response: Considering reviewer ‘s insistence the recommended details are now added in the methodology section. We still believe these details were not necessary and increase the manuscript’s length.
Figure 6 - I appreciate the authors adding confusion matrices to this figure to provide examples of how well the algorithm performs where it was found to work, however, details need to be added to the Methods text about how these validation points were collected and applied to derive these accuracy statistics.
Response: Methodological methods of sample selection for model calibration and validation are now added in the revised manuscript. Thank you.
Reviewer 2 Report
1. The introduction needs to be further condensed, for example, it can be elaborated from the aspects of research background, research progress and the significance of this research to make the introduction more comprehensive.
2. The conclusion can be further condensed according to the research results.
Minor editing of English language required
Author Response
- The introduction needs to be further condensed, for example, it can be elaborated from the aspects of research background, research progress and the significance of this research to make the introduction more comprehensive.
Response: We thank the reviewer for sharing this view. We believe that shortening or condensing the introduction will significantly impact the manuscript quality. The readers will find it hard to follow the methods and result sections. Hence, we did not make any changes.
- The conclusion can be further condensed according to the research results.
Response: We edited the conclusion section to condense it further. Thank you.